# *Dunaliella salina* Alga Protects against Myocardial Ischemia/Reperfusion Injury by Attenuating TLR4 Signaling

**DOI:** 10.3390/ijms24043871

**Published:** 2023-02-15

**Authors:** Chin-Feng Tsai, Hui-Wen Lin, Jiuan-Miaw Liao, Ke-Min Chen, Jen-Wei Tsai, Chia-Sung Chang, Chia-Yu Chou, Hsing-Hui Su, Pei-Hsun Liu, Ya-Chun Chu, Yi-Hsin Wang, Meilin Wang, Shiang-Suo Huang

**Affiliations:** 1Division of Cardiology, Department of Internal Medicine, Chung Shan Medical University Hospital, Taichung 402, Taiwan; 2School of Medicine, Chung Shan Medical University, Taichung 402, Taiwan; 3Department of Optometry, Asia University, Taichung 413, Taiwan; 4Genetics Center, Department of Medical Research, China Medical University Hospital, Taichung 404, Taiwan; 5School of Chinese Medicine, China Medical University, Taichung 404, Taiwan; 6Department of Physiology, School of Medicine, Chung Shan Medical University, Taichung 402, Taiwan; 7Department of Medical Education, Chung Shan Medical University Hospital, Taichung 402, Taiwan; 8Department of Parasitology, School of Medicine, Chung Shan Medical University, Taichung 402, Taiwan; 9Hualien Tzu Chi Hospital, Buddhist Tzu Chi Medical Foundation, Hualien 970, Taiwan; 10Department of Physical Medicine & Rehabilitation, Shin Kong Wu Ho-Su Memorial Hospital, Taipei 111, Taiwan; 11Department of Anesthesiology, Taichung Veterans General Hospital, Taichung 407, Taiwan; 12Department of Pharmacology, School of Medicine, Chung Shan Medical University, Taichung 402, Taiwan; 13Department & Institute of Physiology, National Yang Ming Chiao Tung University, Taipei 112, Taiwan; 14Department of Anesthesiology, Taipei Veterans General Hospital, Taipei 112, Taiwan; 15School of Medicine, National Yang Ming Chiao Tung University, Taipei 112, Taiwan; 16Department of Microbiology and Immunology, School of Medicine, Chung-Shan Medical University, Taichung 402, Taiwan; 17Institute of Medicine, Chung Shan Medical University, Taichung 402, Taiwan; 18Department of Pharmacy, Chung Shan Medical University Hospital, Taichung 402, Taiwan

**Keywords:** *Dunaliella salina*, myocardial ischemia/reperfusion injury, toll-like receptor 4, inflammation, cardioprotection

## Abstract

Myocardial ischemia/reperfusion (I/R) injury is marked by rapid increase in inflammation and not only results in myocardial apoptosis but also compromises the myocardial function. *Dunaliella salina* (*D. salina*), a halophilic unicellular microalga, has been used as a provitamin A carotenoid supplement and color additive. Several studies have reported that *D. salina* extract could attenuate lipopolysaccharides-induced inflammatory effects and regulate the virus-induced inflammatory response in macrophages. However, the effects of *D. salina* on myocardial I/R injury remain unknown. Therefore, we aimed to investigate the cardioprotection of *D. salina* extract in rats subjected to myocardial I/R injury that was induced by occlusion of the left anterior descending coronary artery for 1 h followed by 3 h of reperfusion. Compared with the vehicle group, the myocardial infarct size significantly decreased in rats that were pre-treated with *D. salina*. *D. salina* significantly attenuated the expressions of TLR4, COX-2 and the activity of STAT1, JAK2, IκB, NF-κB. Furthermore, *D. salina* significantly inhibited the activation of caspase-3 and the levels of Beclin-1, p62, LC3-I/II. This study is the first to report that the cardioprotective effects of *D. salina* may mediate anti-inflammatory and anti-apoptotic activities and decrease autophagy through the TLR4-mediated signaling pathway to antagonize myocardial I/R injury.

## 1. Introduction

Clinically, thrombosis or acute alterations in coronary atherosclerotic plaques lead to myocardial ischemia. The most effective strategy to decrease mortality and treat ischemic cardiomyopathy is to restore blood flow as soon as possible. However, several studies have shown that the restoration of blood flow can cause additional heart damage and complications, which can abnormally increase the infarct size; this condition is called myocardial ischemia/reperfusion (I/R) injury [1]. Myocardial I/R injury is the major contributor to the clinical morbidity and mortality associated with coronary artery disease, which is one of the fastest-growing diseases worldwide [2]. Clinically limiting myocardial I/R injury to achieve the best cardioprotective effect is an important strategy for treating myocardial infarction. Increasing evidence has revealed that an intense proinflammatory response triggers myocardial I/R injury because of the reperfusion of blood flow to the coronary artery [3,4]. In addition, studies have demonstrated that chronic inflammation, which is caused by heavy chronic alcohol consumption, long-term smoking or excessive caloric intake, would aggravate the development of myocardial I/R injury [5,6,7]. The inflammatory response to acute myocardial I/R injury plays an important role in exacerbating acute infarcted size and worse prognosis post-I/R injury. Therefore, anti-inflammatory therapy is a critical strategy to protect against myocardial I/R injury [8].

Toll-like receptor 4 (TLR4), a cell surface receptor, is closely related to several inflammatory pathways [9] and regulates apoptosis and autophagy-associated pathogenesis in myocardial I/R injury [10]. A previous study has claimed that the production of inflammatory cytokines at the early stage of myocardial I/R injury depends on TLR4 expression [11]. Cell surface receptor-mediated signaling pathways involving TLR4 and proinflammatory cytokines associated with inflammation as well as myocardial apoptosis and autophagy are the main pathogenic factors in myocardial I/R injury [8]. Therefore, anti-inflammatory and anti-apoptotic activities and the regulation of autophagy may reveal therapeutic options for patients with myocardial I/R injury. As mentioned above, limiting I/R injury-induced TLR4 expression and blocking TLR4-mediated myocardial inflammation, apoptosis and autophagy can confer protection against myocardial I/R injury, which may reduce I/R injury-induced myocardial infarction [12].

Microalgae are considered the richest sources of natural carotenoids, especially strains of the Chlorophyta such as *Dunaliella salina* (*D. salina*) (Chlorophyceae) [13]. *D. salina* is a halophilic unicellular microalga that has a mucus surface coat but no cell wall [14]. It was first found in a sea salt field. In order to survive in this environment, the alga secretes high concentrations of β-carotene for protection against intense light and high concentrations of glycerol for protection against intense osmotic pressure. The notable amounts of carotenoids secreted by *D. salina* mainly include all-trans-β-carotene, 9-cis-β-carotene and 9′-cis-β-carotene [14]. Natural carotenoids have gained increasing attention in recent years because of their health benefits compared to synthetic carotenoids, which are predominantly all-trans compounds and have some controversial benefit [15]. By contrast, intake of food supplements enriched with natural β-carotene containing both cis- and trans- stereoisomers is linked with alleviation of some diseases including atherosclerosis, diabetes and ophthalmologic diseases [16,17]. *D. salina* has been particularly widely studied as it is the richest source of natural β-carotene [18], and long-term study revealed that *D. salina* does not have chronic toxicity in both genders of rodents [19]. In recent years, *D. salina* has been clinically used as a provitamin A supplement and is recommended as a dietary supplement and as a color additive in food and cosmetics, and β-carotene possessing provitamin A activity has been reported to protect against cardiovascular disease [20]. *D. salina* has been reported to improve various diseases related to cardiac dysfunction in obese rats fed a high-fat diet (HFD); this may be related to its anti-oxidant and anti-inflammatory effects [21]. In addition, zeaxanthin heneicosylate isolated from *D. salina* has been demonstrated to ameliorate age-associated cardiac dysfunction in rats by significantly increasing the retinoic acid receptor alpha (RAR-α) expression level and consequently activating retinoid receptors in cardiac tissues [22]. Moreover, *D. salina* attenuates lipopolysaccharides-induced inflammatory effects and regulates the virus-induced inflammatory response in macrophages [23]. *D. salina* is also rich in omega-3 fatty acids. The anti-inflammatory properties of the algal omega-3 fatty acid concentrate may inhibit NF-κB translocation [24]. In addition, *D. salina* extract exerts an anti-inflammatory effect and a high anti-oxidant effect that reduces the virus-induced accumulation of reactive oxygen species and inhibits the synthesis of nitric oxide [25]. Natural products possessing anti-inflammatory and anti-oxidant properties, such as natural carotenoids, natural polyphenolic compound, have significant potential for protecting the heart against myocardial I/R injury [26,27]. However, no study has assessed the potential cardioprotective effect of non-toxicity and β-carotene-rich *D. salina* on myocardial I/R injury. Hence, the present study aimed to assess the possible cardioprotective effect and underlying pharmacological mechanism of *D. salina* in rats subjected to myocardial I/R injury and to determine whether the inhibition of the TLR4 signaling pathway plays a critical role in this effect.

## 2. Results

### 2.1. Myocardial Damage

The infarct size was measured using the Evans blue-TTC double staining method. There were no significant differences in the size of the area at risk among the vehicle, *D. salina*-0.1, and *D. salina*-1 groups, indicating a similar damage after myocardial I/R injury among these groups (Figure 1A). Notably, compared with the vehicle group, the infarct size after myocardial I/R injury decreased in a dose-dependent manner in the *D. salina*-0.1 and *D. salina*-1 groups. Moreover, compared with the vehicle group, the infarct size to area at risk ratio significantly decreased from 28.56% to 16.63% in the *D. salina*-1 group (Figure 1B).

Compared with the vehicle group, the LDH activity (Figure 1C) and troponin I level (Figure 1D) significantly decreased in the carotid blood of the rats in the *D. salina*-1 group after myocardial I/R injury, consistent with the finding that pretreatment with *D. salina* simultaneously decreased the myocardial infarct size; all these factors serve as indicators of cellular damage. These findings indicate the protective effect of 1 mg/kg *D. salina* against myocardial I/R injury.

### 2.2. Hemodynamic Changes during Myocardial I/R Injury

There were no significant differences in the pre-ischemic hemodynamic parameters, including HR, mean BP, ±dp/dt_max_, and LVSP among the three groups of rats. Moreover, there were no significant differences in HR and mean BP between the vehicle- and 1 mg/kg *D. salina*-treated rats during myocardial I/R injury (Figure 2B,C). However, compared with the sham group, the +dP/dt_max_ and LVSP significantly decreased after 1 h of myocardial ischemia and 3 h of reperfusion in the vehicle group. Compared with the vehicle group, ±dP/dt_max_ and LVSP significantly increased in the *D. salina* group (Figure 2D,F). The recovery of ±dP/dt_max_ and LVSP indicated that 1 mg/kg *D. salina* could improve the cardiac functional recovery in rats subjected to myocardial I/R injury.

### 2.3. Effects of D. salina on the Inflammatory Signaling Pathway

The TLR4 signaling pathway plays a central role in myocardial I/R injury and is involved in the inflammatory pathway after I/R injury. As shown in Figure 3A,B, compared with the vehicle group, the expression of TLR4 significantly decreased in the *D. salina*-treated group. This indicates that the cardioprotective effect of *D. salina* may be associated with the TLR4 signaling pathway.

The JAK2–STAT1 pathway is associated with TLR4-mediated inflammatory responses. STAT1 acts as a transcription factor for COX-2; the activation of STAT1 depends on the phosphorylation of JAK2. In the present study, results revealed that compared with the vehicle group, the expression levels of JAK2, p-JAK2, STAT1, and p-STAT1 significantly decreased in the *D. salina*-treated group after myocardial I/R injury. Similarly, The expression level of COX-2 was significantly suppressed by *D. salina* treatment in the rats subjected to myocardial I/R injury. Moreover, myocardial I/R injury induced the phosphorylation of IκB, which led to the translocation of NF-κB to the nucleus and mediated proinflammatory gene expression. We also examined whether *D. salina* could block the activation of IκB to NF-κB during myocardial I/R injury. Compared with the vehicle group, the expression levels of IκB, p-IκB, NF-κB and p-NF-κB significantly decreased in the myocardium of the rats in the *D. salina*-treated group after myocardial I/R injury (Figure 3A,B). We further examined whether the *D. salina*-treated group corresponds to less myocardial inflammation as demonstrated by neutrophil infiltration. We used immunohistochemical staining for MPO on the hearts. We found that administration of D. salina significantly reduced I/R-induced neutrophil infiltration in the myocardium compared with the vehicle group (Figure 3C,D). Thus, *D. salina* may limit myocardial I/R injury-induced inflammation by suppressing not only the JAK2–STAT1 pathway but also the NF-κB-responsive mechanism. 

### 2.4. Effects of D. salina on Apoptosis and Autophagy

Next, we examined whether caspase-3, a key protein in both extrinsic and intrinsic apoptotic pathways, plays a role in the cardioprotective effect of *D. salina* in rats subjected to myocardial I/R injury. Compared with the vehicle group, the expression levels of both procaspase-3 (inactive form) and cleaved caspase-3 (active form) significantly decreased in the *D. salina*-treated group (Figure 4A,B). These findings suggest that *D. salina* attenuates the apoptotic level in the I/R-injured heart.

To detect whether *D. salina* regulates myocardial autophagy after I/R injury, we also examined the expression levels of beclin-1, p62 and LC3. Compared with the vehicle group, the expression levels of beclin-1, p62 and LC3-I/II significantly decreased in the *D. salina*-treated group (Figure 4A,B). These results indicate that *D. salina* decreases myocardial autophagy after I/R injury.

## 3. Discussion

*D. salina* algae have been widely used as health food components, vitamin A precursor supplements, food colorants and food and cosmetics additives. The present study is the first to reveal the cardioprotective effects of *D. salina* in rats subjected to myocardial I/R injury. The main findings of the present study are as follows: (1) Compared with the vehicle, *D. salina* significantly decreased the LDH activity and troponin I level in the plasma and decreased myocardial infarction in the rats after myocardial I/R injury. (2) *D. salina* improved the cardiac functional recovery in the rats subjected to myocardial I/R injury, as evidenced by significantly higher ±dP/dt_max_ and LVSP than those in the vehicle group. (3) *D. salina* significantly attenuated the inflammatory signaling pathway, resulting in decreased expression levels of TLR4, p-JAK2, JAK2, p-STAT1 and STAT1, p-NF-κB, NF-κB, p-IκB, IκB and COX-2. (4) *D. salina* attenuated apoptosis and autophagy by significantly decreasing the expression levels of caspase-3 and those of beclin-1, p62 and LC3-I/II, respectively. These findings suggest that *D. salina* extracts play a role in attenuating TLR4 signaling to inhibit inflammatory responses, apoptosis and autophagy to exert potent cardioprotective activities that protect the myocardium against I/R injury (Figure 5). 

*Dunaliella* strains are well known for being rich in β-carotene [18]. However, excess β-carotene levels may cause a variety of adverse effects, and large-scale prospective randomized trials reported that supplementation with 20 to 30 mg β-carotene per day was associated with an increased risk of lung cancer and cardiovascular disease [28]. Moreover, Csepanyi and colleagues revealed that the low-dose of β-carotene treatment (30 mg/kg/day) significantly reduced the infarcted zone relative to the control group, but this protective effect is abolished in the hearts taken from animals treated with a high-dose of β-carotene (150 mg/kg/day) in the Langendorff model [29]. Recently, they also showed that increasing the concentration of β-carotene does not provide an additional benefit in the diabetic I/R myocardium [30]. *D. salina* has been as well known as the richest source of natural β-carotene, but no study has assessed the potential cardioprotective effect of *D. salina* on myocardial I/R injury. In the present study, our results showed that 1 mg/kg of *D. salina* could improve cardiac functional recovery and significantly decrease the infarct size after acute myocardial I/R injury. The effective dosage of *D. salina* for ameliorating myocardial I/R injury in rats is lower than the dose of β-carotene, which may cause serious side effects in the clinical trials.

Previous reports demonstrated that *D. salina* had good anti-inflammatory activity. El-Baz et al. declared that *D salina* attenuated high-fat diet-induced fibrotic cardiac tissue and congestion of myocardial blood vessels by influencing anti-oxidant and anti-inflammatory effects in obese rats [21]. Moreover, *D. salina* could modulate pseudorabies virus-induced inflammation by inhibition of NF-κB activation via TLR9 dependent-PI3K/Akt inactivation in vitro. In addition, *D. salina* inhibited reactive oxygen species (ROS) accumulation and inflammation by downregulation of STAT-1/3 dependent NF-κB activation in virus-infected RAW264.7 cells [25]. In this study, we found that *D. salina* (10 μM) significantly increased cell viability against OGD/R insult in H9c2 cells, and also significantly inhibited ROS accumulation and inflammatory cytokines emancipation (Appendix A). However, the inflammatory signaling pathway is a principal mediator of myocardial injury during ischemia and reperfusion. The unifying pathophysiological mechanism underlying myocardial I/R injury-induced inflammation could be explained by elevated TLR4 levels after myocardial I/R injury, resulting in the direct activation of the inflammatory pathway [31] to enhance tissue injury. In the present study, we found that compared with the vehicle, *D. salina* significantly decreased the expression level of TLR4. Previous studies have suggested that TLR4 has a direct relationship with the inflammatory signaling pathway (not only the JAK2–STAT1 pathway but also the NF-κB signaling pathway), which finally contributes to the translation of COX-2. The expression levels of many inflammatory cytokines increase after COX-2 upregulation. STAT1 acts as a transcription factor for COX-2; the activation of STAT1 depends on the phosphorylation of JAK2 [32]. NF-kB, an important transcription factor located downstream of the TLR4-mediated signaling pathway, is isolated in the cytoplasm in an inactive form by interacting with IkB under unstimulated conditions. When TLR4 is activated, IκB becomes phosphorylated and degraded, followed by the phosphorylation of NF-κB [33]. Phosphorylated NF-κB enters the nucleus and activates some specific proinflammatory genes, such as COX-2 [34]. Therefore, decreasing the expression level of TLR4 may have an anti-inflammatory effect. We found that compared with the vehicle, *D. salina* significantly decreased the expression levels of several proteins, including JAK2, p-JAK2, STAT1, p-STAT1 and COX-2. Thus, *D. salina* has cardioprotective effects that may involve the inhibition of the TLR4-mediated inflammatory signaling pathway.

In addition, the TLR4 cascade can activate cell apoptosis [35]. Apoptosis is the process of programmed cell death that acts under a physiological mechanism to clean up unwanted cells without triggering an inflammatory response. However, myocardial I/R injury can trigger inappropriate apoptosis and cause worse damage to cardiomyocytes [36]. Moreover, Sheu and colleagues declared that *D. salina* could regulate the antiproliferative effects and induce cell cycle arrest and apoptosis in non-small cell lung cancer cells [37]. In the present study, we assessed the expression levels of procaspase-3 (inactive form) and cleaved caspase-3 (active form) to determine the role of apoptosis in the cardioprotective effect of *D. salina*. We found that compared with the vehicle, *D. salina* significantly decreased the expression levels of procaspase-3 and cleaved caspase-3. Hence, *D. salina* may protect the heart against I/R injury by inhibiting apoptosis.

It is well known that the signaling cascade involved in TLR4-mediated autophagy may play a significant role in the pathogenesis of myocardial I/R injury [10]. Autophagy induces the dysfunction and death of cardiomyocytes, is triggered by myocardial ischemia and is enhanced by reperfusion [38]. However, Beclin-1 is a key factor involved in TLR4-mediated autophagy signaling that can regulate both autophagosome formation and processing [39]. Beclin-1 needs LC3-II as well as p62 for autophagosome formation. Moreover, p62 acts as an autophagy receptor for the degradation of the ubiquitinated substrates. It also facilitates the transport of some degraded organelles and cytosolic proteins to the autophagosome and provides a molecular link between autophagy and ubiquitination [40]. LC3-I, the cytosolic form, is cleaved to form LC3-II during autophagosome formation. Furthermore, using cross species RNA-seq meta-analysis and machine-learning models identified that microalga Dunaliella could regulate chaperone-mediated autophagy, the lipid and nitrogen metabolism and ROS scavenging-related genes against salt stress conditions [41]. In the present study, we found that pretreatment with *D. salina* significantly decreased the expression levels of Beclin-1, p62, LC3-I and LC3-II in rat heart tissue after myocardial I/R injury. These results suggest that the cardioprotective effect of *D. salina* is associated with the anti-autophagic pathway.

In conclusion, *D. salina* has been clinically used as a dietary provitamin A carotenoid supplement in recent years. The present study is the first to demonstrate that pretreatment with *D. salina* could protect the myocardium against I/R injury. We speculate that the beneficial cardioprotective effects of *D. salina* are mediated by the attenuation of the TLR4-mediated signaling pathway and play a role in attenuating inflammation, apoptosis, and autophagy in rats subjected to myocardial I/R injury. Our results suggest that a dietary supplement of *D. salina* might be beneficial for protecting the heart against myocardial I/R injury.

## 4. Materials and Methods

### 4.1. Animals

The protocols of the present study conformed to those published in the Guide for the Care and Use of Laboratory Animals by the National Research Council of the National Academies (NIH publication, revised 2011). Six-week-old male Sprague–Dawley rats (LASCO Co., Taipei, Taiwan) were used in the study and were housed in the Animal Center of Chung Shan Medical University at a stable temperature of 25 ± 1 °C and humidity of 55 ± 5% under a 12 h light–dark cycle. The rats were fed normal chow and given water ad libitum. The surgical procedures for inducing myocardial I/R injury were performed under supervision and approved by the Institutional Animal Care and Use Committee of Chung Shan Medical University, Taichung, Taiwan (IACUC 1924). Every effort was made to minimize animal suffering and to reduce the number of rats sacrificed.

### 4.2. Preparation of D. salina Extract

The dried powder of *D. salina* cultivated in Taiwan was obtained from Gong Bih Enterprise Co., Ltd. (Wunlin, Taiwan). *D. salina* extract was prepared according to the method of Lin et al. [25]. Extract the algal sample (10 g) in 250 mL hexane/acetone/ethanol (2:1:1, *v*/*v*/*v*) with a shaker for 2 h at room temperature. Then, saponification was performed by adding 10 mL of 40% methanolic KOH at 25 °C for 16 h. The solvent was evaporated to dryness to yield carotenoid extract. The carotenoids composition in the extract was estimated by high-performance liquid chromatography (HPLC) under the conditions determined in the previous report [25]: column, YMC C30 (250 × 4.6 mm, 5 μm) (Waters Co., Milford, MA, USA); mobile phase, methanol (MeOH)–acetonitrile–H2O (84/14/2, *v*/*v*/*v*)/CH2Cl2 = 75/25 (*v*/*v*); flow rate, 1 mL/min; detection, 210–650 nm at a rate of 1.00 spectrum/s. The equipment consisted of a PrimeLine Gradient Model 500G HPLC pump system (Analytical Scientific Instruments, Inc., El Sobrante, CA, USA) and an S-3210 photodiode-array detector (Schambeck SFD GmbH, Bad Honnef, Germany). Moreover, all-trans-β-carotene and 9- or 9′-cis-β-carotene accounted for 92.46% of the major components of the *D. salina* extract [14]. The extracts was then dissolved in dimethyl sulfoxide (DMSO) for the experiment. 

### 4.3. D. salina Administration

The *D. salina* extract was freshly prepared before administration. The experimental rats were randomly divided into four groups: (1) sham: rats subjected to all surgical procedures, except for left anterior descending (LAD) coronary artery occlusion, (2) vehicle: rats treated with vehicle (0.1% DMSO in normal saline), (3) *D. salina*-0.1: rats treated with 0.1 mg/kg *D. salina* extract, and (4) *D. salina*-1: rats treated with 1 mg/kg *D. salina* extract. The *D. salina* extract or vehicle was intravenously administered 15 min before LAD coronary artery occlusion. Our preliminary results found that 1 mg/kg of *D. salina* can effectively protect the heart against myocardial I/R injury. In order to reduce the number of animals used in the present study, the dose used to investigate the mechanisms of *D. salina* cardioprotection was tested from 1 mg/kg. This dose is lower than the long-term safe dose of 100 mg/kg, which is administrated in both genders of rats [19].

### 4.4. Surgical Protocol

Myocardial I/R injury was induced by temporary occlusion of the LAD coronary artery, as described previously [42]. Briefly, the rats were initially anesthetized with a single intraperitoneal injection of urethane (1.25 g/kg). The jugular vein and femoral artery were cannulated to administer drugs and to continuously monitor the blood pressure (BP) and heart rate [39] during the experiment. Left thoracotomy cut the fourth and fifth ribs to approximately 2 mm of the sternum in order to expose the heart; a 6/0 silk suture was then used to surround the LAD coronary artery. The tightening of the suture occluded the LAD coronary artery for 1 h and the tension of the suture was then released to cause reperfusion for 3 h. 

### 4.5. Evaluation of Cardiac Function

A Millar catheter was inserted into the left ventricular cavity via the right common carotid artery and changes in the left ventricular systolic pressure (LVSP), left ventricular diastolic pressure (LVDP) and maximal slope of systolic pressure increment (+dP/dt_max_) and diastolic decrement (−dP/dt_max_) were continuously recorded using a Transonic Scisense Pressure Measurement system (SP200, Transonic Scisense Inc., London, ON, Canada).

### 4.6. Measurement of Myocardial Injury and Collection of Myocardial Samples

Myocardial infarct size was determined by a double staining technique using Evans blue and 2,3,5-triphenyltetrazolium chloride (TTC; Sigma-Aldrich, St. Louis, MO, USA). At the end of the experiment, the coronary artery was re-occluded and 0.1% Evans blue solution was injected intravenously, and the non-ischemic myocardium was stained to determine the area at risk. Hearts were sectioned transversely into 2 mm thick slices using a cardiac slicer matrix (Jacobowitz Systems, Zivic-Miller Laboratories Inc., Allison Park, PA, USA). Heart sections were stained with 2% TTC for 30 min at 37 °C in the dark. Place the sections in a solution containing 10% formalin for 1 day at room temperature. Infarcted tissue sections were scanned and tissue weight was assessed by distinguishing the normal myocardium (stained blue) from the area at risk and infarct area (unstained) in a TTC staining assay. Arterial blood collected from the carotid catheter in rats that survived after 60 min of ischemia and 3 h of reperfusion was centrifuged at 3000× *g* for 10 min to separate plasma and determine myocardial damage. Myocardial cellular damage was determined using automated clinical analyzers to measure plasma activities of lactate dehydrogenase [43] (ADVIA 1800, Siemens Healthcare Diagnostics Inc., Tarrytown, NY, USA) and the levels of Troponin-I (Centaur, Siemens Healthcare Diagnostics Inc., NY, USA).

### 4.7. Protein Extraction and Western Blot Analysis

In order to reduce the number of animals used in the present study, 1 mg/kg *D. salina* was used as the effective dose to investigate the mechanism underlying the cardioprotective effect of *D. salina*. Protein expression was evaluated by western blot as previously described procedures [44]. The primary antibodies against COX-2 (Cayman Chemical, Ann Arbor, MI, USA) included TLR4, phospho-JAK2 (p-JAK2), JAK2 (Santa Cruz, Dallas, TX, USA), phospho-NF-κB (p-NF-κB), NF-κB, phospho-IκB (p-IκB), IκB, procaspase-3, cleaved caspase-3 (Cell Signaling, Danvers, MA, USA), signal transducer and activator of transcription-1 (STAT1), phospho-STAT1 (p-STAT1) (Abcam, Cambridge, UK), light chain 3 (LC3), p62 and Beclin-1 (Novus, Centennial, CO, USA). β-actin (Abcam, Cambridge, UK) was used as an internal loading control. The membranes were incubated with an HRP-conjugated secondary antibody (Jackson Immuno Research Laboratories, West Grove, PA, USA) prior to chemiluminescence detection (Thermo Scientific, Waltham, MA, USA).

### 4.8. Immunohistochemical Staining 

Rat hearts were individually fixed in 10% neutral buffered formalin for 24 h. They were then dehydrated in a graded ethanol series (50%, 75%, 95% and 100%), cleared in xylene, and embedded in paraffin at 55 °C for 24 h. For immunohistochemical analysis, serial sections of relatively thick (5 μm) wax-embedded hearts were cut, mounted on glass slides, deparaffinized and then rehydrated in the same manner as for histological analysis. Sections were then treated with 3% H_2_O_2_ in methanol for 10 min to inactivate any endogenous peroxidases and washed 3 times with PBS for 5 min each. Block with 3% BSA at room temperature for 1 h, then add 1:50 diluted rabbit anti-human MPO polyclonal antibody and incubate at 37 °C for 1 h. After 3 additional washes in PBS, sections were incubated with HRP-conjugated rabbit anti-goat IgG diluted 1:100 in 1% BSA for 1 h at 37 °C, followed by 3 additional washes in PBS. Finally, sections were incubated with 3,3’-diaminobenzidine (0.3 mg/mL) in 100 mM Tris (pH 7.5) containing 0.3 μL H_2_O_2_/_mL_ for 3 min at room temperature. After three washes in PBS, the sections were fixed in 50% glycerol in PBS and examined under a light microscope.

### 4.9. Statistical Analysis

The data are expressed as the mean ± standard error of the mean. Statistical analyses of differences were performed using one-way analysis of variance for combined data, followed by Bonferroni tests. *p* < 0.05 was considered statistically significant.

## Figures and Tables

**Figure 1 ijms-24-03871-f001:**
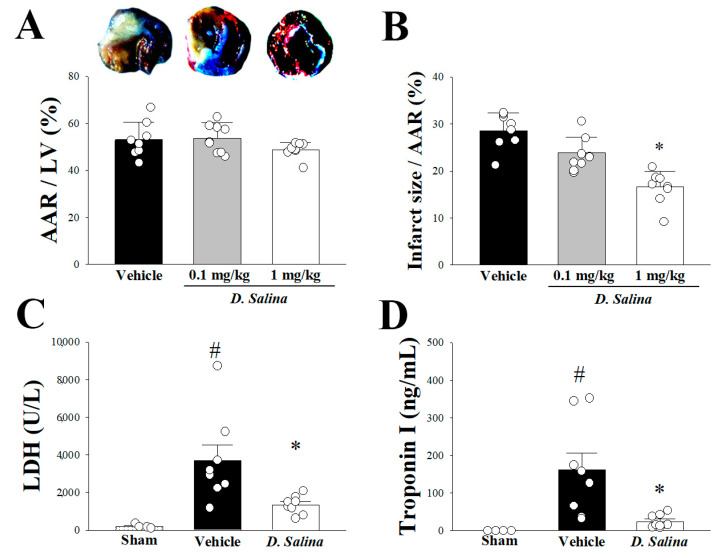
Effect of *D. salina* on I/R induced myocardial damage in rats. (**A**) Representative Evans blue/TTC staining of heart sections from vehicle-treated group and different dose (0.1 and 1 mg/kg) of *D. salina* groups after 1 h of LAD coronary artery occlusion and 3 h of reperfusion. Perfused area is seen in blue; area at risk (AAR) is seen in red and white. Infarction is seen in white. Quantitative AAR/Ventricle % (ratio of non-blue area to total area), (**B**) the risk zone infarcted % (ratio of white area to AAR), (**C**) plasma lactate dehydrogenase activity and (**D**) plasma troponin-I level, in vehicle and 1 mg/kg *D. salina*-treated rats. The results are shown as mean ± S.E.M. (*n* = 4–9); # *p* < 0.05 compared with the sham group; * *p* < 0.05 compared with the vehicle-treated group.

**Figure 2 ijms-24-03871-f002:**
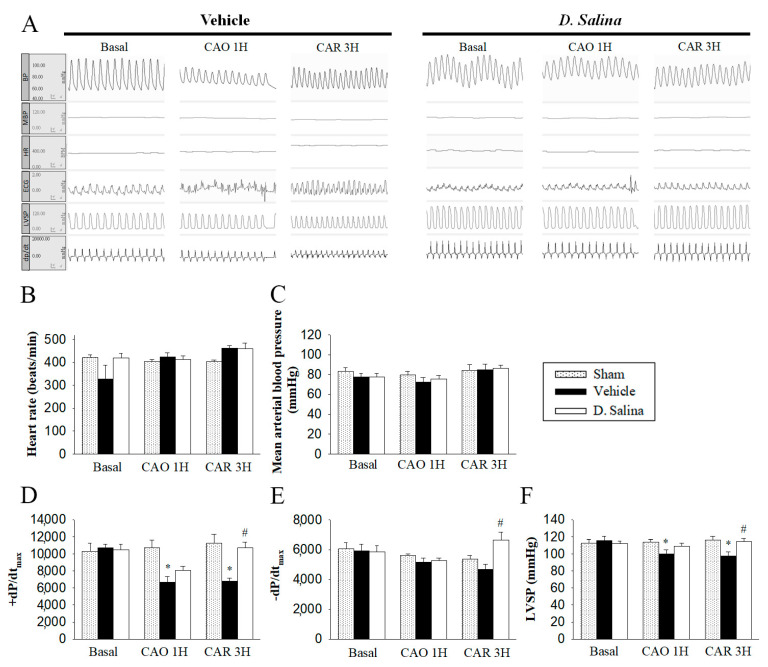
Effect of *D. salina* on hemodynamic parameters in rats subjected to myocardial I/R injury. (**A**) Representative image of hemodynamic parameters between the vehicle- and 1 mg/kg *D. salina*-treated rats during myocardial I/R injury. Quantitatively analyzed results of (**B**) heart rate, (**C**) mean arterial blood pressure, (**D**) maximum rates of pressure change in the LV (+dP/dt), (**E**) minimum rates of pressure change in the LV (−dP/dt) and (**F**) left ventricular systolic pressure (LVSP) were recorded by physiological parameters (CAO1H, 1 h after LAD coronary artery occlusion; CAR3H, 3 h after LAD coronary artery reperfusion). Values are mean ± S.E.M.; (*n* = 8 per group); # *p* < 0.05 compared with the sham group; * *p* < 0.05 compared with the vehicle-treated group.

**Figure 3 ijms-24-03871-f003:**
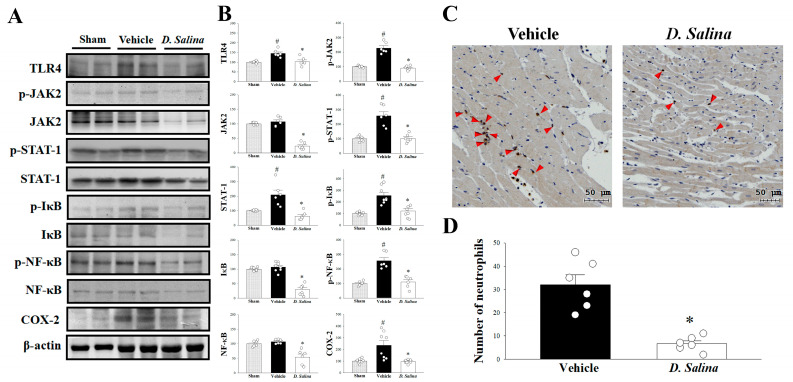
*D. salina* deceased inflammation after myocardial I/R injury. (**A**) Representative images of the Western blot results in heart tissue. (**B**) Graphs represent the quantitative differences of TLR4, p-JAK2 and JAK2, p-STAT1 and STAT1, p-IκB and IκB, p-NF-κB and NF-κB, COX-2 between sham, vehicle and 1 mg/kg *D. salina*-treated groups in myocardial I/R animals. β-actin was used as a loading control for the blots. (**C**) Representative areas of MPO immunohistochemical staining slides of rats subjected to myocardial I/R injury. The red arrow indicates the infiltration of neutrophils. (**D**) Quantitative analysis of MPO positive cells of rat heart slides that were subjected to myocardial I/R injury. Data are normalized with the mean expression from the sham group. # *p* < 0.05 compared with the sham group; * *p* < 0.05 compared with the vehicle-treated group. Values are expressed as mean ± S.E.M. (*n* = 6–8).

**Figure 4 ijms-24-03871-f004:**
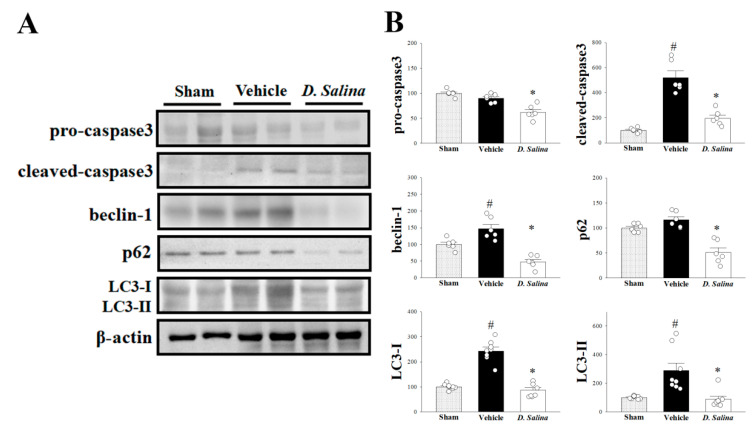
*D. salina* decreased apoptosis and autophagy after myocardial I/R injury. (**A**) Representative images of the Western blot results in heart tissue. (**B**) Graphs represent the quantitative differences of pro-caspase3 and cleaved-caspase3, beclin-1, p62, LC3-I and LC3-II between sham, vehicle and 1 mg/kg *D. salina*-treated groups in myocardial I/R animals. β-actin was used as a loading control for the blots. Data are normalized with the mean expression from the sham group. # *p* < 0.05 compared with the sham group; * *p* < 0.05 compared with the vehicle group. Values are expressed as mean ± S.E.M. (*n* = 6–8).

**Figure 5 ijms-24-03871-f005:**
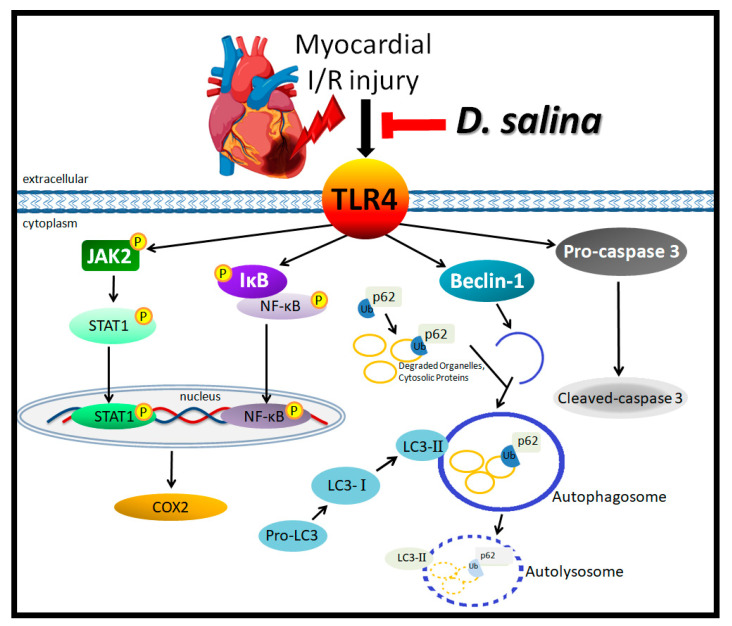
Schematic diagram of the effects of *D. salina* treatment regulates TLR4 signaling against myocardial I/R injury. Pre-treatment with *D. salina* mediates anti-inflammatory and anti-apoptotic activities and decreases autophagy through the TLR4-mediated signaling pathway to antagonize myocardial I/R injury.

## Data Availability

The data presented in this study are available on request from the corresponding author.

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
