# Peer review of "Dunaliella salina Alga Protects against Myocardial Ischemia/Reperfusion Injury by Attenuating TLR4 Signaling"

_ijms, 2023, doi:10.3390/ijms24043871_

Round 1
Reviewer 1 Report
In this manuscript, Tsai et al presented results indicating that extracts from D Salina alga protect against cardiac I/R injury by targeting TLR4 signaling. D salina alga extracts showed beneficial effects when administrated before LAD occlusion. However, whether the extracts can still protect against cardiac I/R injury administrated post-occlusion is unknown.
Below are the concerns the authors need to address.
1. In Figure 1A, which dose was used to for the D salina-treated rats? Why basal level LVSP and dp/dt values are significantly higher in the treated rats compared to the vehicle treated ones?
2. In Figure 2A, the border between red/blue/white area is not clearly identified due to weak contrast.
3. In Figure 3A, p-I kappa B protein band sizes are different between vehicle and D salina treated rats.
4. It looks like D. salina treatment induced a overall reduction in the expression of proteins in inflammatory, apoptosis, and autophagy pathways. Detection of protein expression level in the sham group is recommended.
Author Response
- In Figure 1A, which dose was used to for the D salina-treated rats? Why basal level LVSP and dp/dt values are significantly higher in the treated rats compared to the vehicle treated ones?
Response: Thanks for the question. We examined the hemodynamic changes between the sham, vehicle- and 1 mg/kg salina-treated rats before and after 1 h of myocardial ischemia and 3 h of reperfusion. There were no significant differences in the basal level hemodynamic parameters, including HR, mean BP, ±dp/dtmax, and LVSP, among the three groups of rats. - In Figure 2A, the border between red/blue/white area is not clearly identified due to weak contrast.
Response: Thanks for your suggestion. We have increased the resolution and contrast of the Evans blue/TTC staining images to ensure that clearly displayed comparative results in Figure 1A (Figure 2A in the previous edition). - In Figure 3A, p-I kappa B protein band sizes are different between vehicle and D salina treated rats.
Response: Thanks for your question. We have corrected the band of p-I kappa B protein in Figure 3A. - It looks like D. salina treatment induced a overall reduction in the expression of proteins in inflammatory, apoptosis, and autophagy pathways. Detection of protein expression level in the sham group is recommended.
Response: Thanks for your suggestion. We have followed the reviewer’s suggestion and detected the protein expression in the sham group.
Reviewer 2 Report
The paper by Tsai et al., entitled “D.salina alga protects against myocardial ischemia/reperfusion injury by attenuating TLR4 signaling” showed that D.salina extract bears cardioprotective properties in I/R injury and is mediated via TLR4 signaling. Although authors performed comprehensive in vivo study, research lacks key fundamental in vitro experiments that can really conclude that the process is driven by TLR4 signaling. To consider publication in this paper authors should address this concerns:
Major points:
-As stated above; a comprehensive in vivo study was carried out with very demanding surgical animal model, still paper lacks cell experiments to really show that D.salina extract has cardioprotective role in I/R injury; perhaps some experiments on rat cardiomyocytes (e.g. H9C2 cells)?
-the authors should perform also in vitro experiments on TLR4 knock out cells (e.g.CRISPR/Cas generated) to really determine that process is driven via TLR4
-the authors should also check if the ROS formation (due to I/R) are hampered by D.salina treatment
-in all western blot pictures only two bands are presented from some groups, I would like to see the results from all animals and also from all the experimental groups (not only vehicle and D.salina; also from sham animals and also from animals that were untouched-meaning no surgical procedure was performed on them); that would importantly show the role of D.salina
-Figure 1A should be more legible; again what is with other animals, here only one animal is shown (the same goes for Figure 2A)
- Figure 1B: the figure legend does not contain the information regarding the dose of the extract, given to the animals, of which results are presented here; also what is with animals (measurements) that were given lower dose of D.salina
-Figure 2D-E; again which animals (what dosage) are presented here; authors should show all experimental animals (from both dosages as in Fig.2B-C)
-in all cases (for all figures) I would really like to see graphs were individual points (corresponding to each animal) are shown
-the authors should also determine pro- and antiinflammatory cytokines in the blood of the animals; also some histology slides of the hearts with specific cell staining would be beneficial (e.g. Ly6G, CD86 etc.)
Minor comments:
-in figure legends the dosage of D.salina is missing
- page 4, line 24; authors should be more specific (describe them) about the methods chosen to determine the state of the injury
-page 3, line 46; cite the preliminary study
Author Response
Major points:
-As stated above; a comprehensive in vivo study was carried out with very demanding surgical animal model, still paper lacks cell experiments to really show that D.salina extract has cardioprotective role in I/R injury; perhaps some experiments on rat cardiomyocytes (e.g. H9C2 cells)?
Response: Thanks for your suggestion. We have followed the reviewer’s suggestion and established the oxygen-glucose deprivation and reoxygenation (OGD/R) model in H9c2 cells to investigate the cardioprotection of D.salina extract. We found that D. salina significantly increased the cell viability. The result have been provided to the supplementary data (Figure S1), and also have been added to the Discussion section in red.
-the authors should perform also in vitro experiments on TLR4 knock out cells (e.g.CRISPR/Cas generated) to really determine that process is driven via TLR4
Response: Thanks for your suggestion. In our study, we found that D. salina decreased the expression of TLR4, so we did not use TLR4 knockout cells for further experiments.
-the authors should also check if the ROS formation (due to I/R) are hampered by D.salina treatment
Response: Thanks for your suggestion. In order to reduce animal sacrifice, we have established the oxygen-glucose deprivation and reoxygenation (OGD/R) model in H9c2 cells to mimic the in vivo model of myocardial I/R injury. We found that D. salina significantly decreased the ROS accumulation in H9c2 cells after OGD/R injury. The result have been provided to the supplementary data (Figure S1), and also have been added to the Discussion section in red.
-in all western blot pictures only two bands are presented from some groups, I would like to see the results from all animals and also from all the experimental groups (not only vehicle and D.salina; also from sham animals and also from animals that were untouched-meaning no surgical procedure was performed on them); that would importantly show the role of D.salina
Response: Thanks for your suggestion. We have followed the reviewer’s suggestion and detected the protein expression in the sham group.
-Figure 1A should be more legible; again what is with other animals, here only one animal is shown (the same goes for Figure 2A)
Response: Thanks for your suggestion. We have added the dot graphs to show the results of each animal. Figure 1A and Figure 2A are schematic diagrams representing Evans blue/TTC staining of heart sections and images of hemodynamic parameters.
- Figure 1B: the figure legend does not contain the information regarding the dose of the extract, given to the animals, of which results are presented here; also what is with animals (measurements) that were given lower dose of D.salina
Response: Thanks for your suggestion. We have added the information regarding the dose of the extract in the figure legend.
-Figure 2D-E; again which animals (what dosage) are presented here; authors should show all experimental animals (from both dosages as in Fig.2B-C)
Response: Thanks for your suggestion. We have added the information regarding the dose of the extract in the figure legend.
-in all cases (for all figures) I would really like to see graphs were individual points (corresponding to each animal) are shown
Response: Thanks for your suggestion. We have added the dot graphs to show the individual results.
-the authors should also determine pro- and antiinflammatory cytokines in the blood of the animals; also some histology slides of the hearts with specific cell staining would be beneficial (e.g. Ly6G, CD86 etc.)
Response: Thanks for your suggestion. We further examined whether the D. salina-treated group corresponds to less myocardial inflammation as demonstrated by neutrophil infiltration. We used immunohistochemical staining for MPO on the hearts. We found that administration of D. salina significantly reduced I/R-induced neutrophil infiltration in the myocardium compared with the vehicle group. The data has been added in Figure 3C-D and a description has been added in the Result section in red. Furthermore, we also found that D. salina significantly decreased the TNF-a and IL-1b levels after OGD/R injury in H9c2 cells compared with the control group. These results have been presented in the supplementary data (Figure S1) and added in the Discussion section in red.
Minor comments:
-in figure legends the dosage of D.salina is missing
Response: Thanks for your suggestion. We have added the information regarding the dose of the extract in the figure legend.
- page 4, line 24; authors should be more specific (describe them) about the methods chosen to determine the state of the injury
Response: Thanks for your suggestion. We have clearly described the measurement of myocardial injury in the Materials and Methods (2.6 Measurement of myocardial injury and collection of myocardial samples).
-page 3, line 46; cite the preliminary study
Response: Thanks for your advice. Sorry, we meant to present the preliminary result rather than the preliminary study, and we have corrected the description.
Round 2
Reviewer 2 Report
The authors made all the needed corrections.